

# The South-American distribution and southernmost record of *Biomphalaria peregrina*—a potential intermediate host of schistosomiasis

Alejandra Rumi[1,2], Roberto Eugenio Vogler[1,2,3,*] and Ariel Aníbal Beltramino[1,2,4,*]

[1] División Zoología Invertebrados, Facultad de Ciencias Naturales y Museo, Universidad Nacional de La Plata, La Plata, Buenos Aires, Argentina

[2] Consejo Nacional de Investigaciones Científicas y Técnicas (CONICET), CABA, Argentina

[3] Instituto de Biología Subtropical, Universidad Nacional de Misiones- Consejo Nacional de Investigaciones Científicas y Técnicas (CONICET), Posadas, Misiones, Argentina

[4] Departamento de Biología, Facultad de Ciencias Exactas, Químicas y Naturales, Universidad Nacional de Misiones, Posadas, Misiones, Argentina

[*] These authors contributed equally to this work.

Corresponding author
Alejandra Rumi,
alerumi@fcnym.unlp.edu.ar

## ABSTRACT

Schistosomiasis remains a major parasitic disease, endemic in large parts of South America. Five neotropical species of *Biomphalaria* have been found to act as intermediate hosts of *Schistosoma mansoni* in natural populations, while others have been shown to be susceptible in experimental infections, although not found infected in the field. Among these potential intermediate hosts, *Biomphalaria peregrina* represents the most widespread species in South America, with confirmed occurrence records from Venezuela to northern Patagonia. In this study, we report the southernmost record for the species at the Pinturas River, in southern Patagonia, which finding implies a southward reassessment of the limit for the known species of this genus. The identities of the individuals from this population were confirmed through morphological examination, and by means of two mitochondrial genes, *cytochrome oxidase subunit I (COI)* and *16S-rRNA*. With both markers, phylogenetic analyses were conducted in order to compare the genetic background of individuals from the Pinturas River with previously genetically characterized strains of *B. peregrina* from various South-American locations. In addition, we produced a potential distribution model of *B. peregrina* in South America and identified the environmental variables that best predict that distribution. The model was estimated through a maximum entropy algorithm and run with occurrence points obtained from several sources, including the scientific literature and international databases, along with climatic and hydrographic variables. Different phylogenetic analyses with either the *COI* or *16S-rRNA* sequences did not conflict, but rather gave very similar topological organizations. Two major groups were identified, with sequences from the Pinturas River grouping together with haplotypes from subtropical and temperate regions. The model developed had a satisfactory performance for the study area. We observed that the areas with higher habitat suitability were found to be mainly linked to subtropical and temperate regions of South America between 15° and 45° south latitude, with different moderate- and low-suitability areas outside this range. We also identified the coldest temperatures as the main predictors of the potential distribution of this snail. Susceptibility surveys would

be required to evaluate if southern populations of *B. peregrina* still retain their potential as intermediate hosts of *S. mansoni*.

## INTRODUCTION

Schistosomiasis is an acute and chronic parasitic disease that affects at least 258 million people worldwide. Seventy-eight countries are considered endemic for schistosomiasis, with the populations of 52 countries requiring preventive chemotherapy (*World Health Organization, 2015*). There are two major forms of schistosomiasis—intestinal and urogenital—caused by five main species of trematodes, and the disease is frequent in tropical and subtropical regions, particularly in poor communities (*World Health Organization, 2015*). In the Americas, the only species present is *Schistosoma mansoni* (Sambon, 1907) (Digenea); which species is associated with intestinal schistosomiasisis, and is transmitted by the freshwater snails of the genus *Biomphalaria* Preston, 1910; e.g., *Biomphalaria glabrata* (Say, 1818), *Biomphalaria tenagophila* (*D'Orbigny, 1835*), and *Biomphalaria straminea* (*Dunker, 1848*). Other snail species, such as *Biomphalaria peregrina* (*D'Orbigny, 1835*), have been infected experimentally and are thus considered as potential hosts of *S. mansoni* (*Paraense & Côrrea, 1973*).

In the Americas, schistosomiasis currently occurs in Brazil, Venezuela, Surinam, Puerto Rico, the Dominican Republic, and on several islands of the Lesser Antilles, with recent evidence indicating a spread from northeastern Brazil southward (*Spatz et al., 2000*; *Kloos et al., 2008*; *Noya et al., 2015*). An expansion of the current disease-distribution area can be expected, since the geographical range of snails that can act as intermedial hosts (IHs) is wider than that of the pathogen (*Pan American Health Organization, 2010*). The southern area with the highest risk of establishing an endemic and a new focus of disease is located in the northeast of Argentina (the NEA Region), where the majority of the *Biomphalaria* species inhabit the major rivers of the Del-Plata basin. Five of those species are listed as potential IHs of schistosomiasis: *B. tenagophila*, *B. straminea*, *B. peregrina*, *B. orbignyi* Paraense, 1975, and *B. oligoza Paraense, 1974* (*Rumi, 1991*; *Rumi & Vogler, 2014*).

The identification and recognition of *Biomphalaria* species thus far has mainly relied on features of shell morphology, radula, jaw and the reproductive system (*cf. Paraense, 1966*; *Paraense, 1975a*; *Paraense, 2003*; *Rumi, 1991*). Character similarity among the species, however, has in fact hampered classification, as the case of *B. peregrina, B. orbignyi* and *B. oligoza* found in Argentina whose morphological similarities difficult correct identification (*Paraense, 1988*; *Spatz et al., 2000*; *Vidigal et al., 2000a*; *Estrada et al., 2006*). Within the historical context, several of the South-American species were described in genera of doubtful taxonomic position, as the example of *Taphius* Adams & Adams, 1855; *Biomphalaria* Preston, 1910; *Tropicorbis* Brown & Pilsbry, 1914; *Platytaphius* Pilsbry, 1924; and *Australorbis* Pilsbry, 1934. In addition, the original diagnoses were mostly made based

only on shell characters. Both situations facilitated the generation of species of doubtful validity. Subsequently, the anatomical evidence has demonstrated that no differences really existed between the genera and the various taxa all belonging to the same genus (*Paraense, 1958*). Although the oldest name was *Taphius*, in 1965 the International Committee of Zoological Nomenclature imposed the sole name *Biomphalaria*, considering that one to be the most widespread in the world (*Barbosa et al., 1961*; *Paraense, 2008*). More recently, though, difficulties concerning morphological identification have been overcome through the use of molecular-genetic techniques that have contributed to delimiting species, mainly those occurring in the Neotropics (e.g., *Caldeira et al., 1998*; *Vidigal et al., 1998*; *Vidigal et al., 2004*; *Spatz et al., 2000*).

Among the *Biomphalaria* species in South America, *B. peregrina* exhibits one of the most widespread distributions—and one involving a great diversity of hydrologic systems—with that species thus far having been recorded from Venezuela to northern Patagonia, Argentina (*Paraense, 1966*; *Paraense, 2003*; *Paraense, 2004*; *Rumi, 1991*; *Spatz et al., 2000*; *Núñez, Gutiérrez Gregoric & Rumi, 2010*; *Standley et al., 2011*). In the study reported here, the presence of *B. peregrina* in southern Patagonia is now documented for the first time, that location being the southernmost record for the species—and the genus as well—worldwide. In order to confirm the identity of this most southerly population, we assessed the main conchological and anatomical diagnostic characters (e.g., shell, genitalia, and radula) and obtained DNA sequences of the mitochondrial *cytochrome oxidase subunit I* (*COI*) and the *16S-rRNA* genes. Based on both markers, we compared the genetic background of individuals from the Pinturas River with previously genetically characterized strains of *B. peregrina* from various South-American locations and examined the intra-specific phylogenetic position of the recently discovered population. In addition, upon consideration that *B. peregrina* represents a potential host for schistosomiasis, we produced a predictive model of the species's spatial distribution in South America and identified the environmental variables that best predict its location. The resulting model indicating the likely whereabouts of that potential host will hopefully provide further guidance for future efforts aimed at schistosomiasis surveillance and control.

## MATERIALS AND METHODS

The material analyzed (19 specimens) came from the Pinturas River, Santa Cruz province, Argentina (Gatherer: Hugo Merlo Álvarez, col. Date: 15-XI-2013, geographical coordinates: $46°50'6''S; 70°27'38''W$; 355 m above sea level). The specimens studied were deposited in the malacological collections at the Museo de La Plata (DZI-MLP-Ma), Buenos Aires province (MLP-Ma No 14186).

### Morphological examination

The adult specimens ($n = 12$; MLP-Ma No 14186/1–14186/12) were analyzed according to *Paraense & Deslandes (1956)*, *Paraense (1966)*, *Paraense (1974)*, *Paraense (1975a)*, *Paraense (1975b)* and *Rumi (1991)*. The morphology of the shell, the radula, and the reproductive system were analyzed. The soft parts were separated from the shell for subsequent processing and fixed in Railliet–Henry —93% (v/v) distilled water, 2% (v/v) glacial acetic acid, 5%

(v/v) formaldehyde, and 6 g sodium chloride per liter—or 90% (v/v) aqueous alcohol. Shell measurements (maximum and minimum diameter and height) were obtained with a Mitutoyo digital calipter. The dissection was done under a stereoscopic binocular microscope (LEICA MZ6) (individuals MLP-Ma No 14186/1–14186/8).

The radula and jaw were cleaned following a modification of the non-destructive method described by *Holznagel (1998)* and *Vogler et al. (2016)*: the structures were separated from the mass of tissue and placed in 1.5-ml microtubes containing 500 µl NET buffer (1 ml 1M Tris pH 8.0, 2 ml 0.5 M ethylenediaminetetraacetic acid, 1 ml 5 M NaCl, 20 ml 10% (w/v) sodium dodecyl sulfate, 76 ml water) and 10 µl of Proteinase K (20 mg/ml) were added. The samples were then incubated at 37 °C with a subsequent renewal of the NET buffer and Proteinase K to verify the absence of tissue. After two washes with distilled water, 25% (v/v) aqueous ethanol was added for preservation. Finally the radula and jaw were examined by scanning electron microscopy (JEOL 6360) in the Museum of La Plata (individuals MLP-Ma No 14186/1, MLP-Ma No 14186/2, MLP-Ma No 14186/4 and MLP-Ma No 14186/6). The radular formula gives the number of teeth per row: ((number of left teeth) + (number of central teeth) + (number of right teeth)) plus the number of transverse rows (*Paraense & Deslandes, 1956*).

## DNA extraction, polymerase chain reaction (PCR) and DNA sequencing

Total genomic DNA was extracted from the foot muscle of five individual snails (MLP-Ma No 14186/1–14186/5) through the use of the DNeasy Blood & Tissue kit (Qiagen, Valencia, CA, USA) according to the manufacturer's protocol. Partial sequences of the mitochondrial *cytochrome oxidase subunit I (COI)* and the *16S-rRNA* genes were amplified by means of the primers LCO1490 (5′–GGT CAA CAA ATC ATA AAG ATA TTG G–3′) and HCO2198 (5′–TAA ACT TCA GGG TGA CCA AAA AAT CA–3′) for *COI* (*Folmer et al., 1994*), and 16SF-104 (5′–GAC TGT GCT AAG GTA GCA TAA T–3′) and 16SR-472 (5′–TCG TAG TCC AAC ATC GAG GTC A–3′) for *16S-rRNA* (*Ramírez & Ramírez, 2010*). The amplification of the *COI* region was conducted following *Vogler et al. (2014)*. The amplification of *16S-rRNA* was performed in a total volume of 30 µl containing 30–50 ng of template DNA, 0.2 µM of each primer, 1X PCR green buffer, 0.2 mM dNTPs, and 1 U Dream *Taq* DNA Polymerase (Thermo Scientific, Waltham, MA, USA). The thermocycling profile was 35 cycles of 30 s at 94 °C, 30 s at 48 °C, 1 min at 72 °C followed by a final extension of 1 min at 72 °C. The amplifications were run on a T18 thermocycler (Ivema Desarrollos). The PCR products were purified by means of a ADN PuriPrep-GP kit (InBio-Highway, Tandil, Buenos Aires). After purification, both DNA strands for each gene were then directly cycle-sequenced (Macrogen Inc., Seoul, South Korea). The resulting sequences were trimmed to remove the primers, and the consensus sequences between forward and reverse sequencing were obtained by means of the BioEdit 7.2.5 software (*Hall, 1999*).

## Phylogenetic analysis

The consensus sequences of the individuals were compared to reference sequences in GenBank through the use of the BLASTN algorithm (*Altschul et al., 1990*) to identify

similar sequences in order to confirm the morphology-based identification. Subsequently, phylogenetic analyses were conducted to explore the possible intra-specific phylogenetic relationships of the genetic sequences from the southernmost individuals from the Pinturas River to those of other *B. peregrina* individuals from various locations available in GenBank (Table 1). Sequences of other similar species to *B. peregrina* were not used in the analysis. The phylogenetic analyses were carried out separately for each mitochondrial region as follows: the sequence alignment was performed with Clustal X 2.1 (*Larkin et al., 2007*), and optimized by visual inspection. The total lengths of the matrices analyzed were 546 bp for the *COI* gene, and 269 bp for the *16S-rRNA* locus. The data were subjected to four different phylogenetic analyses by the methods of neighbor joining (NJ), maximum parsimony (MP), maximum likelihood (ML), and Bayesian inference (BI). The NJ analysis was conducted with MEGA 6.06 (*Tamura et al., 2013*) through the use of the Kimura's two-parameter (K2P) substitution model. The MP and ML analyses were carried out with PAUP*4.0b10 (*Swofford, 2002*). The MP was conducted by means of a heuristic search with the characters equally weighted, tree bisection and reconnection branch-swapping, and 10 random stepwise additions. The optimal model of nucleotide substitution for ML inference was evaluated by the likelihood-ratio test and selected by means of the corrected Akaike Information Criterion with Jmodeltest 2.1.7 (*Darriba et al., 2012*). The TVM+I (for *COI*), and the TIM1+I (for *16S-rRNA*) substitution models were used as evolutionary paradigms. Statistical support for the resulting phylogenies was assessed by the bootstrap method with 1,000 replicates (*Felsenstein, 1985*). BI was performed with Mr. Bayes 3.2.6 (*Ronquist et al., 2012*). Two runs were conducted simultaneously with 4 Markov chains that went for $10^6$ generations, sampling every 100 generations. The first 10,001 generations of each run were discarded as burn-in, and the remaining 18,000 trees were used to estimate posterior probabilities. In addition, the number of haplotypes in the dataset was explored with DnaSP 5.10 (*Librado & Rozas, 2009*) and the genetic distances estimated in MEGA 6.06 through the use of the number of differences ($p$) and the K2P-substitution model. Since we obtained shorter *16S-rRNA* sequences than previously reported for *B. peregrina* from Argentina that did not include previously characterized molecular diversity (*Standley et al., 2011*), the *COI* data were employed only for estimating the number of haplotypes and genetic distances.

## Species-distribution model

The study area comprised all of the South-American countries. The occurrence data for *B. peregrina* were retrieved from the scientific literature, and from malacological collections and international databases (Table 2). All together, 689 spatially unique records were used. When the coordinates of localities were lacking, those data were derived secondarily following *Wieczorek, Guo & Hijmans (2004)*. Twenty-three environmental variables were used as predictors; comprising 19 climatic, three hydrologic, and one topographical (Table 3). The variables were downloaded from WorldClim (http://www.worldclim.org) and HydroSHEDS (http://hydrosheds.cr.usgs.gov) at a spatial resolution of 30 arc seconds (∼1 km$^2$). WorldClim and HydroSHEDS provide climatic information derived from weather stations spanning 1950–2000 and hydrographic data obtained from a STRM

**Table 1  Information on the samples used in the phylogenetic reconstruction of *Biomphalaria peregrina*.**

| Sample | Geographical origin | Voucher # | GenBank accession # COI | 16S-rRNA | Reference |
|---|---|---|---|---|---|
| *B. tenagophila*[b] | Mogi das Cruzes, São Paulo, Brazil | LBMSU547 | KF926202 | KF892001 | R Tuan, MCA Guimaraes, FPO Ohlweiler & RGS Palasio (2013, unpublished data)[a] |
| | | | | | R Tuan & RGS Palasio (2013, unpublished data)[a] |
| *B. pfeifferi*[b] | Abu Usher, Sudan | – | DQ084835 | DQ084857 | *Jørgensen, Kristensen & Stothard (2007)* |
| *B. peregrina* | | | | | |
| | Pinturas River, Santa Cruz, Argentina | MLP-Ma14186 | KY124272 | KY124273 | This work |
| | Agua Escondida, Mendoza, Argentina | – | GU168593 | GU168591 | *Standley et al. (2011)* |
| | | | | GU168592 | |
| | La Plata, Buenos Aires, Argentina | UCH La Plata1 | JN621901 | JF309030 | *Collado, Vila & Méndez (2011)* |
| | | UCH La Plata2 | JN621902 | JF309031 | *Collado & Méndez (2012)* |
| | | UCH La Plata3 | JN621903 | JF309032 | |
| | Rancharia, São Paulo, Brazil | LBMSU584 | KF926176 | – | R Tuan, MCA Guimaraes, FPO Ohlweiler & RGS Palasio (2013, unpublished data)[a] |
| | Bagé, Rio Grande do Sul, Brazil | LBMSU663 | KX354439 | – | RGS Palasio & R Tuan (2016, unpublished data)[a] |
| | Ipaussu, São Paulo, Brazil | LBMSU761 | KX354440 | – | RGS Palasio & R Tuan (2016, unpublished data)[a] |
| | | LBMSU756 | KX354441 | – | RGS Palasio & R Tuan (2016, unpublished data)[a] |
| | | LBMSU755 | KX354442 | – | RGS Palasio & R Tuan (2016, unpublished data)[a] |
| | | LBMSU338 | – | KF892035 | R Tuan & RGS Palasio (2013, unpublished data)[a] |
| | Ourinhos, São Paulo, Brazil | LBMSU747 | KX354443 | – | RGS Palasio & R Tuan (2016, unpublished data)[a] |
| | | LBMSU739 | KX354444 | – | RGS Palasio & R Tuan (2016, unpublished data)[a] |
| | | LBMSU300 | – | KF892034 | R Tuan & RGS Palasio (2013, unpublished data)[a] |
| | Martinópolis, São Paulo, Brazil | LBMSU582 | KF926180 | – | R Tuan, MCA Guimaraes, FPO Ohlweiler & RGS Palasio (2013, unpublished data)[a] |
| | | LBMSU581 | KX354445 | KF892036 | RGS Palasio & R Tuan (2016, unpublished data)[a] |
| | | | | | R Tuan & RGS Palasio (2013, unpublished data)[a] |
| | Nova Lima, Minas Gerais, Brazil | – | – | AY030232 | *DeJong et al. (2001)* |
| | San Antonio, Uruguay | – | – | AY030231 | *DeJong et al. (2001)* |

**Notes.**
[a]GenBank unpublished sequences: the sequence author and submission year are indicated.
[b]Outgroup species.
LBMSU,  Laboratório de Bioquímica e Biologia Molecular, Superintendência de Controle de Endemias do Estado de São Paulo, Brazil; MLP,  Museo de La Plata, Argentina; UCH,  Universidad de Chile, Chile.

**Table 2** Sources of *Biomphalaria peregrina* occurrences in South America used in the distribution model.

| Country | Occurrences | Sources consulted[a] |
|---|---|---|
| Argentina | 343 | *D'Orbigny (1835)*, *Paraense (1966)*, *Bonetto, Rumi & Tassara (1990)*, *Castellanos & Miquel (1991)*, *Rumi (1991)*, *Rumi et al. (1996)*, *Rumi, Tassara & Bonetto (1997)*, *Flores & Brugni (2005)*, *Rumi et al. (2006)*, *Rumi et al. (2008)*, *Ciocco & Scheibler (2008)*, *Standley et al. (2011)*<br>*Malacological collections*: CECOAL; IFML; FIOCRUZ; MACN; MLP |
| Bolivia | 8 | *Paraense (1966)*<br>*Malacological collections*: MACN |
| Brazil | 241 | *Paraense (1966)*, *Teles, Pereira & Richinitti (1991)*, *Prando & Bacha (1995)*, *DeJong et al. (2001)*, *Pepe et al. (2009)*<br>*Malacological collections*: FIOCRUZ<br>*Websites*: GenBank; WMSDB |
| Chile | 24 | *Dunker (1848)*, *Biese (1951)*, *Barbosa, Coelho & Carneiro (1956)*<br>*Malacological collections*: FIOCRUZ; MACN |
| Colombia | 1 | *Website*: WMSDB |
| Ecuador | 12 | *D'Orbigny (1835)*, *Cousin (1887)*, *Paraense (1966)*, *Paraense (2004)*<br>*Malacological collections*: FIOCRUZ<br>*Website*: WMSDB |
| Paraguay | 29 | In *Quintana (1982)*, *Paravicini (1894)*, *Bertoni (1925)*, *Schade (1965)*, *Russell (1972)*, *Moreno González (1981)*<br>*Malacological collections*: MACN |
| Peru | 6 | *Paraense (2003)*<br>*Malacological collections*: FIOCRUZ<br>*Website*: WMSDB |
| Uruguay | 23 | *Paraense (1966)*, *DeJong et al. (2001)*, *Scarabino (2004)*<br>*Malacological collections*: FIOCRUZ; MACN<br>*Website*: GanBank; WMSDB |
| Venezuela | 2 | *Website*: WMSDB |

**Notes.**

[a]CECOAL, Centro de Ecología Aplicada del Litoral; FIOCRUZ, Fundação Oswaldo Cruz; IFML, Instituto Fundación Miguel Lillo; MACN, Museo Argentino de Ciencias Naturales; MLP, Museo de La Plata; WMSDB, Worldwide Mollusc Species Data Base.

digital-elevation model, respectively (*Hijmans et al., 2005*; *Lehner, Verdin & Jarvis, 2008*). These variables have been commonly employed for generating distribution models in gastropods, including freshwater members, such as those belonging to the genus *Biomphalaria* (*Scholte et al., 2012*; *Vogler et al., 2013*; *Pedersen et al., 2014*; *Beltramino et al., 2015*; *Martín, Ovando & Seuffert, 2016*). All environmental layers were trimmed to the study area. The potential-distribution model was estimated by using a maximum entropy algorithm in MaxEnt 3.3.3k (*Phillips, Anderson & Schapire, 2006*; *Phillips & Dudík, 2008*). The data were randomly divided into the training data (75% of occurrences) and the model-testing data (the remaining 25%). The output was computed as *logistic*, which

**Table 3  Variables used in the model development.** Temperatures are expressed in °C*10, precipitations in mm, elevation above sea level in m, and flow accumulation in number of cells.

| Variable | Description |
| --- | --- |
| alt | Altitude |
| bio1 | Annual mean temperature |
| bio2 | Mean diurnal range (monthly mean, T° max-T° min) |
| bio3 | Isothermality (bio2/bio7) × 100 |
| bio4 | Temperature seasonality (standard deviation × 100) |
| bio5 | Maximum temperature of warmest month |
| bio6 | Minimum temperature of coldest month |
| bio7 | Temperature annual range (bio5-bio6) |
| bio8 | Mean temperature of wettest quarter |
| bio9 | Mean temperature of driest quarter |
| bio10 | Mean temperature of the warmest quarter |
| bio11 | Mean temperature of coldest quarter |
| bio12 | Annual precipitation |
| bio13 | Precipitation of wettest month |
| bio14 | Precipitation of driest month |
| bio15 | Precipitation seasonality (coefficient of variation) |
| bio16 | Precipitation of wettest quarter |
| bio17 | Precipitation of driest quarter |
| bio18 | Precipitation of the warmest quarter |
| bio19 | Precipitation of the coldest quarter |
| acc | Flow accumulation |
| dir | Flow direction |
| con | Hydrologically conditioned elevation |

setting returns a map with an estimated probability ranging between 0 (no probability of the species presence) and 1 (high probability of presence). The resulting model was assessed by estimating the area under the receiver-operating-characteristic curve (ROC-curve analyses; *Fielding & Bell, 1997*). The relative contribution of variables to the development of the model was evaluated by means of a jackknife test and through the response curves obtained in MaxEnt following *Meichtry de Zaburlín et al. (2016)*.

## RESULTS

### Morphological examination

**Shell** (Fig. 1A). The empty shell is fundamentally light brown in color, with the growth lines clearly visible. The whorls, up to $5\frac{3}{4}$ in number, increase slowly and display a rounded surface on both sides. The shells exhibit a more or less marked deflection of the outer whorl to the left. The maximum value for the larger diameter was 13.5 mm (mean = 9.77 mm, SD = 1.54 mm, $n = 12$); the maximum value for the smaller diameter was 8.98 mm (mean = 7.53 mm, SD = 1.40 mm, $n = 11$). The greatest height has been 4.6 mm (mean = 3.86 mm, SD = 0.64 mm, $n = 12$).

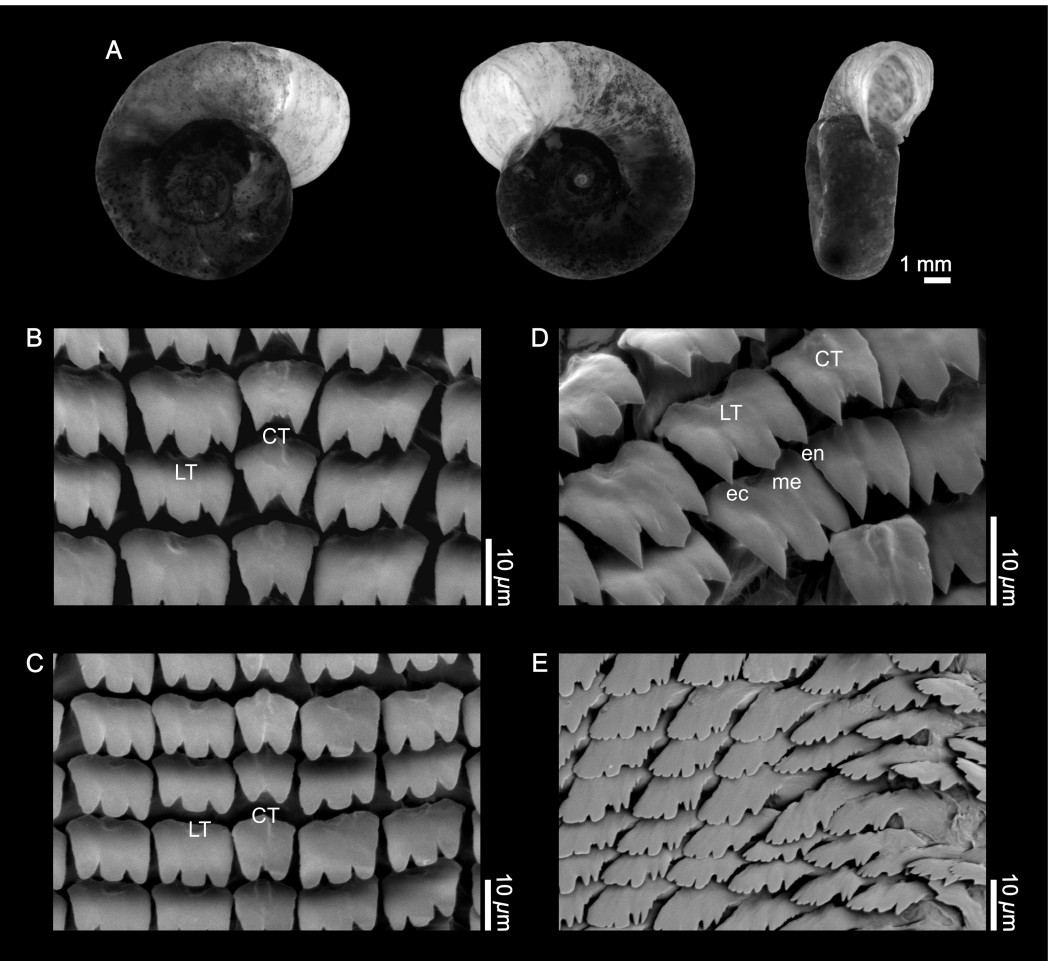

**Figure 1** **External shell morphology and radula of *Biomphalaria peregrina* from the Pinturas River, Argentina.** (A) right, left, and ventral views. (B–D) detail of the rachidian or central (CT) and lateral teeth (LT); ec, ectocone; en, endocone; me, mesocone. (E) detail of marginal teeth.

**Radula** (Figs. 1B–1E). The central tooth is rather asymmetrical, bicuspid, with or without accessory cusps, with the base without special features. There are 106 rows of teeth; with 8 laterals and 12 marginals per half row. The first lateral tooth is tricuspid with the mesocone more developed, and with the free border rounded or in the shape of a sword point. A crest with a central depression toward the posterior part of the tooth is evident. Finally, the marginal teeth are without special features. Radular formula: [20-1-20] 106.

**Jaw.** The jaw is T-shaped, with the dorsal part composed of a single crescent-shaped piece. The features correspond to the standard description of the species and do not differ from those of other *Biomphalaria* species.

**Genital system.** The specimens showed the typology described for *B. peregrina*, without any special variations as ilustrated in *Rumi (1991)*. The following distinctive characters are highlighted: in the male genital system, the vas deferens was wider distally and narrower than the the middle portion of the penis sheath; the number of the prostatic diverticula was 12–14. In the female system, the apical portion of the spermathecal body was almost

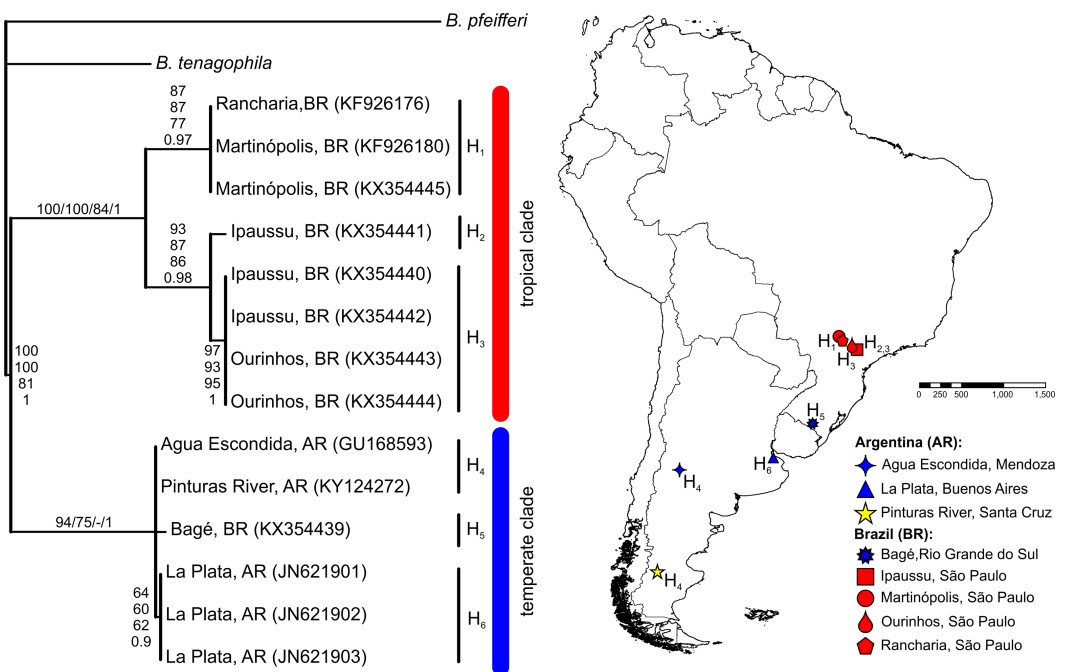

**Figure 2** **Bayesian tree of *Biomphalaria peregrina* based on the partial *COI* gene.** The bootstrap values for the NJ, MP, ML trees and posterior-probability values for BI are shown above and below the branches. The numbers within parentheses are GenBank-accession numbers. The geographical distribution of the localities sampled and the haplotypes (H) is shown. The literature references to the sequences are given in Table 1.

always covered by an anterior and long prostatic diverticulum; the vaginal pouch was well developed.

## Phylogenetic analysis

The BLASTN search results, with the obtained partial *COI* and *16S-rRNA* sequences from the Pinturas River as the query sequences, showed top-ranking scores and a 100% sequence identity with sequences available in the GenBank from Agua Escondida, Mendoza (Argentina) and confirmed their identity as *B. peregrina*. Partial DNA sequences consisted of 655 bp for *COI* and 265 bp for *16S-rRNA*. Both markers contained no variation among the five individuals sequenced, resulting in the existence of only one haplotype per marker. After the inclusion of GenBank sequences from other locations with subsequent alignment, six unique haplotypes were identified within the *COI* dataset (Fig. 2). The sequence divergence among haplotypes is presented in Table 4. Different phylogenetic analyses with either the *COI* or *16S-rRNA* marker did not conflict; rather, both loci gave very similar topological organizations for the NJ, MP, and BI trees with minor differences in the ML-tree organization. In all instances, two major groups were identified, referred to as the *tropical* and *temperate* clades (Figs. 2 and 3). In terms of that subdivision, the specimens from the Pinturas River were placed within the temperate group.

**Table 4  Genetic distances among *COI* haplotypes of *Biomphalaria peregrina*.** The distances are listed as uncorrected (below the diagonal) and corrected by the Kimura's two parameter substitution model (above the diagonal).

| | H₁ | H₂ | H₃ | H₄ | H₅ | H₆ | GenBank accession numbers[a] |
|---|---|---|---|---|---|---|---|
| H₁ | – | 0.012987 | 0.012987 | 0.026321 | 0.030198 | 0.028256 | H₁: KF926176; KF926180; KX354445 |
| H₂ | 0.012820 | – | 0.011116 | 0.032147 | 0.036068 | 0.034104 | H₂: KX354441 |
| H₃ | 0.012820 | 0.010989 | – | 0.032147 | 0.036068 | 0.034104 | H₃: KX354440; KX354442; KX354443; KX354444 |
| H₄ | 0.025641 | 0.031135 | 0.031135 | – | 0.003676 | 0.001834 | H₄: KY124272; GU168593 |
| H₅ | 0.029304 | 0.034798 | 0.034798 | 0.003663 | – | 0.005525 | H₅: KX354439 |
| H₆ | 0.027472 | 0.032967 | 0.032967 | 0.001831 | 0.005494 | – | H₆: JN621901; JN621902; JN621903 |

**Notes.**
  [a]References to the sequences are provided in Table 1.

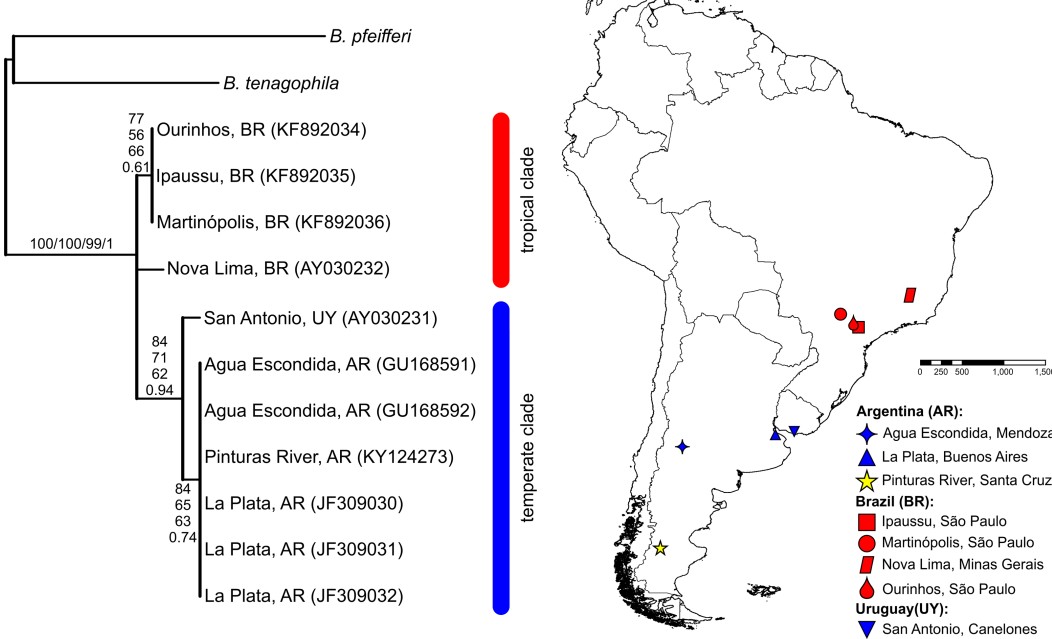

**Figure 3  Bayesian tree of *Biomphalaria peregrina* based on the partial *16S-rRNA* gene.** The bootstrap values for the NJ, MP, ML trees and posterior-probability values for BI are shown above and below the branches. The numbers within parentheses are GenBank-accession numbers. The geographical distribution of the localities sampled is shown. The literature references to the sequences are given in Table 1.

## Species-distribution model

Figure 4 illustrates the potential distribution area for *B. peregrina*. The model conformed well to expectations, with values for the area under the curve of 0.927 for the training data and 0.905 for the test data, with a standard deviation of 0.009. The areas with higher probability of the snail's presence were found to be mainly linked to subtropical and temperate regions of South America between 15° and 45° south latitude, comprising central and northeastern Argentina, central Chile, eastern Paraguay, southeastern Brazil, and southern Bolivia and Uruguay. In addition, regions with a moderate to high habitat suitability were predicted for Peru, Ecuador, Colombia, and Venezuela. The areas of lower habitat suitability were located in French Guiana, Guyana, Suriname, Venezuela,

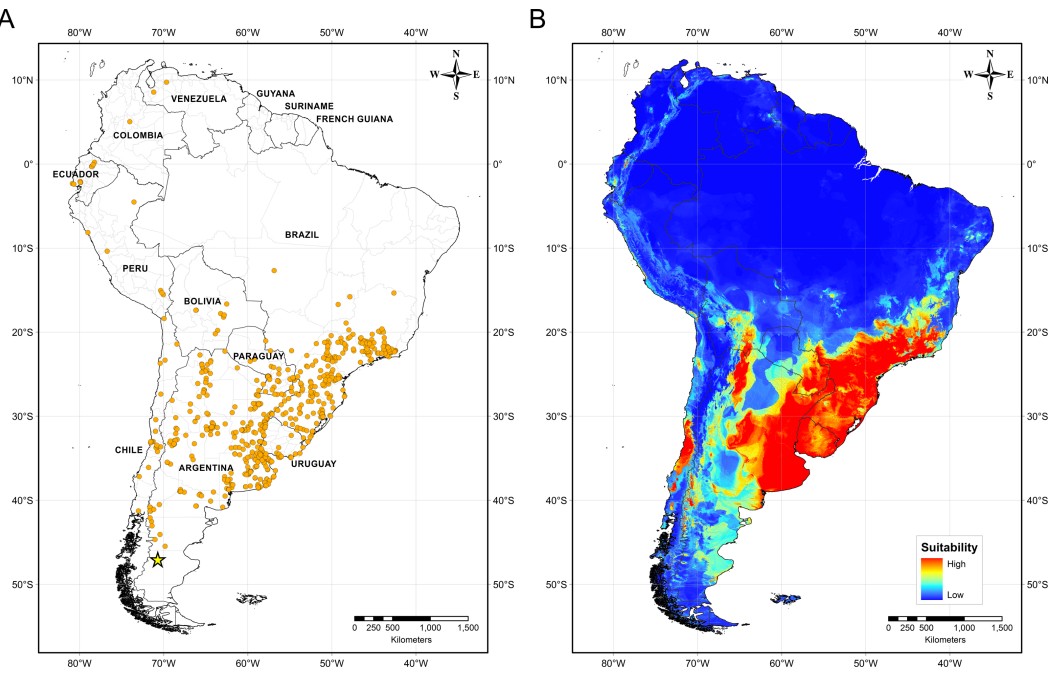

**Figure 4** **Distribution of *Biomphalaria peregrina* in South America.** (A) Records of the snail's presence used in the modelling approach, with the southernmost record being from the Pinturas River, Argentina, as indicated by a yellow star. (B) Potential distribution in logistic format. The color code for location suitability and thus probability of the snail's presence: red, very high; yellow, high; azure, moderate; blue, low.

and a large area of Brazil (Fig. 4B). The jackknife test showed that the mean temperature of the coldest quarter (bio11), the minimum temperature of coldest month (bio6) and the annual mean temperature (bio1) were the variables that most greatly influenced the model development when used in isolation (Fig. 5A). The flow accumulation produced a reduction in training gain when removed from the model, thus indicating that that variable contained information necessary for the model. The remaining predictors contributed less to the modelling. Figure 5B, contains the marginal-response curves for the four strongest environmental predictors—i.e., the mean temperature of the coldest quarter, the minimum temperature of the coldest month, the annual mean temperature, and the flow accumulation.

## DISCUSSION

The conchology and anatomy of the reproductive system of the specimens from the Pinturas River were consistent with the descriptions of *Paraense & Deslandes (1956)* and *Paraense (1966)* for *B. peregrina*. The shells of the individuals from that river recall the original descriptions by the first authors for *Australorbis inflexus* (nowadays considered synonymous with *B. peregrina*; *Paraense, 1966*) from Pouso Alegre, Minas Gerais, Brazil because of the strong inflection of the aperture located toward the left side. Nevertheless, the specimens from the Pinturas River have much wider and rounder shells on both sides than those described for *A. inflexus*. The radulae of the individuals from south Patagonia

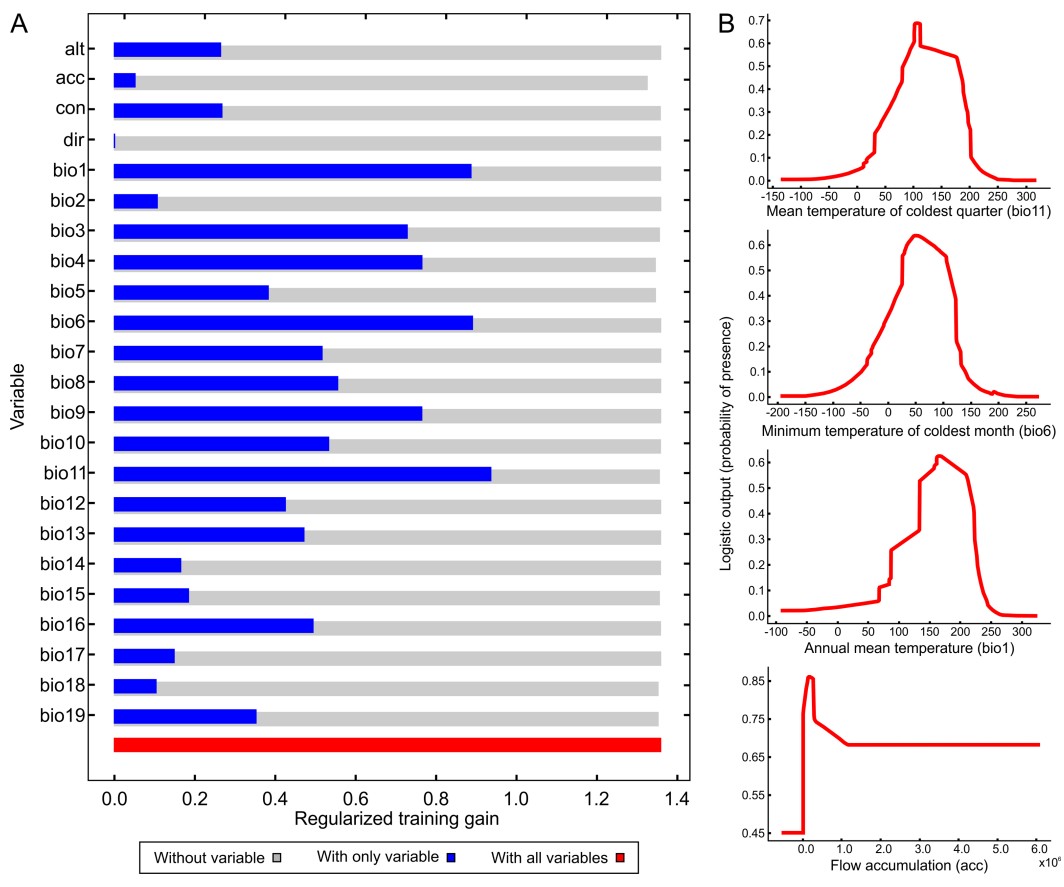

**Figure 5** Relative influence of the environmental variables for the potential distribution of *Biomphalaria peregrina* in South America. (A) Jackknife test determining the contribution of each environmental variable to the development of the model. In the figure, the regularized training gain is plotted on the abscissa for each of the variables indicated on the ordinate. Color code: gray, without a variable; blue, with only a single variable; red, with all variables. (B) Marginal-response curves for the four strongest environmental predictors. In each of the figures, the logistic output, a measure of the probability of presence, is plotted on the ordinate for—from the upper to the lower figure—the mean temperature of the coldest quarter (bio11), the minimum temperature of the coldest month (bio6), the annual mean temperature (bio1), and the flow accumulation (acc).

have a folding and a groove between the cuspids of the central tooth that has not been described for other populations of this species. As mentioned in the Results section, the anatomy of the male genitalia of the specimens examined here exhibited a vas deferens wider distally and narrower than the middle portion of the penis sheath, which is typical of *B. peregrina*. This feature allowed distinguishing the individuals of the Pinturas River from the similar species *B. orbignyi*, whose widest portion of the distal segment of the vas deferens is nearly equal in width to the penis sheath (*Paraense, 1975b*). On the other hand, the number of prostatic diverticula in the material examined allowed distinguishing from *B. oligoza* whose diverticula number is much smaller (typically 1–6, rarely 7), and are also poorly developed. Other morphological features which confirmed the specific identity of the specimens studied as *B. peregrina* were the apical portion of the spermathecal body—almost always covered by an anterior and long prostatic diverticulum—and the vaginal pouch

well developed (*Paraense, 1966*; *Paraense, 1975a*). When analyzing individuals collected at Agua Escondida, Mendoza, Argentina, *Standley et al. (2011)* found a conchology and reproductive organ morphology consistent with that of typical *B. peregrina*, as found here for the Pinturas River individuals. However, those authors stated that "the internal morphology of *B. peregrina* has been shown to distinguish it even from closely related species, but not from *B. orbignyi* (*Spatz et al., 2000*)", and consequently they did not rule out *B. orbignyi* as an alternative for the identity of the individuals from that location. This is not the case for the individuals of the Pinturas River, given that the statment in *Standley et al. (2011)* is not necessarily true, as the differentiation of *B. peregrina* and *B. orbignyi* rest on differences in the wider portion of the distal segment of the vas deferens (*Paraense, 1975b*).

In addition, the morphologically identified specimens from the Pinturas River were confirmed genetically as being *B. peregrina* and a single haplotype was found among the five individuals analysed for each mitochondrial-marker loci examined. Likewise, *Standley et al. (2011)* reported the absence of variation in the *COI* marker for the population studied from Agua Escondida, Mendoza, Argentina. As mentioned above, despite having found the typical internal morphology of *B. peregrina*, those authors did not rule out *B. orbignyi* as a possible alternative, although the similar species was not included in that genetic approach. Those authors also suggested that the lack of variation in that population could be owing to a founding event in recent years since the species of *Biomphalaria* are hermaphroditic and capable of rapidly colonizing a new locality, even from a sole individual founder. Strikingly, a single *COI* haplotype seems be shared between the populations of Agua Escondida and the Pinturas River, which are geographically distant (roughly around 1,200 km). Two main (non-exclusive) alternatives might explain such pattern: (i) the resolution of the *COI* sequence used is not resolutive enough to detect genetic variability and discriminate these samples, and (ii) these populations share a close common evolutionary history. Further studies based on a greater number of specimens and molecular markers would be required to test these alternatives.

The phylogenetic trees revealed that *B. peregrina* can be considered as divided into two clades, here referred to as the tropical and the temperate. For the *COI* marker, the first clade comprised exclusively individuals from Brazil, whereas the second included specimens from southern Brazil and Argentina. For the *16S-rRNA*, the sequences from Brazil all grouped together within the tropical clade, whereas the temperate clade included only sequences from Argentina and Uruguay. These clades could be linked to the biogeography and ecologic history of *B. peregrina* in terms of the colonization of freshwater environments. Nonetheless, of pertinence to emphasize here is that although *B. peregrina* possesses a wide distribution in South America, at the present time only few DNA sequences are available in GenBank for this snail, as well as for the morphologically similar species *B. oligoza* and *B. orbignyi*. Early molecular studies based on the internal transcribed spacer region of the ribosomal DNA sequences (*ITS*) demonstrated a closer relationship between *B. peregrina* and *B. oligoza* (*Spatz et al., 2000*; *Vidigal et al., 2000b*), but still further research is required to fully understand the relationship among the similar species (*B. peregrina*, *B. oligoza*, and *B. orbignyi*) and to verify whether other molecular markers such

as *COI*—currently unavailable for *B. oligoza* and *B. orbignyi*—corroborates the classical morphological identification. Therefore, investigations focussing on the phylogeography of *B. peregrina* are required to acquire a comprehensive understanding of its evolutionary history, similar to those carried out for *B. glabrata* (*Mavárez et al., 2002*; *DeJong et al., 2003*), especially since *B. peregrina* has been shown to occupy a basal position in the phylogenetic analyses of the genus (*DeJong et al., 2001*; *Jarne, Pointier & David, 2011*).

*Biomphalaria peregrina,* as its name might suggest—from the Latin *peregrinus*, meaning wanderer, in reference to its wide geographical distribution (*Paraense, 1975a*)—is the species most widely distributed in South America within the genus. The occurrence of representative individuals from Venezuela down to one of the most southern areas of Patagonia in the Province of Santa Cruz demonstrates that the species possesses an ample range of environmental tolerance. Even so, the historical registration of *B. peregrina* and the area of its potential distribution described here nevertheless indicate that the species's greatest abundance and dispersion are within the Atlantic corridor in the south of Brazil as well as in the northeast and pampean region of Argentina. According to the results obtained here, and in agreement with the findings of *Rumi (1991)*, the potential distribution of *B. peregrina* involves the most southerly areas of the Great-Del-Plata basin, then spreads to the west through the endorrheic basins of Córdoba—the ancient beds of the Paraná River—toward the Andes region, and finally, remaining east of the Andes, extends to the north of the continent through regions which correspond to specific altitude areas within the Andes, up to Venezuela, where the registers become quite scarce. To the south, the distribution stretches from the pampas of the Buenos Aires province, which area corresponds to the central-Argentine malacological region V (*Núñez, Gutiérrez Gregoric & Rumi, 2010*), to the west and to the south, occupying areas on both sides of the Andes range. On the Argentine side, *B. peregrina* inhabits the malacological regions VI in Cuyo, VII in northern Patagonia, and to the south reaches the region VIII in southern Patagonia (*Núñez, Gutiérrez Gregoric & Rumi, 2010*). On the Chilean side of the Andes, the species is dispersed from the region IX to IV.

Each species of *Biomphalaria* that is a natural or potential host for *S. mansoni* consists in populations that exhibit varying degrees of susceptibility to different local strains of the parasite (*Paraense & Côrrea, 1973*; *Paraense & Côrrea, 1978*; *Paraense & Côrrea, 1985*; *Coelho et al., 2004*; *Simões et al., 2013*; *Marques et al., 2014*). The susceptibility to *S. mansoni* has been shown to be heritable and linked to the gene pool of the IHs, each of which—according to its capability for reproduction through cross- or self-fertilization—produces progeny with differing degrees of parasitotrophic susceptibility (*Newton, 1953*; *Richards & Merritt Jr, 1972*; *Richards, 1973*; *Richards, 1975*; *Coelho et al., 2004*). As mentioned above in the Results section under the investigation of the genetic background of *B. peregrina*, the haplotypes were found to be bifurcated into two, those from tropical areas and those inhabiting the southern cone of South America, where the climate goes from subtropical to temperate in the Patagonian region to the south. These results are highly relevant since *Paraense & Côrrea (1973)* demonstrated experimentally that populations of *B. peregrina* from Lapa of Paraná (in Brazil), and Chillogallo (in Ecuador) are markedly susceptible to infection with the BH (Belo-Horizonte) and SJ (San-Jose) strains of *S. mansoni*, though

these *B. peregrina* strains have not yet been found to be infected in the wild. Although the susceptibility to different strains of *S. mansoni* of populations of *B. peregrina* that inhabit subtropical or temperate areas have still not been evaluated, other *Biomphalaria* species such as *B. tenagophila* and *B. straminea* in the northeast of Argentina have indeed been found to serve experimentally as IHs of several strains of *S. mansoni* (*Borda & Rea, 1997*; *Simões et al., 2013*). In this context, new studies on the susceptibility and genetic variation of diverse variants and morphotypes of *B. peregrina* are needed involving populations that contain the haplotypes identified here as subtropical and temperate. For example, those individuals characterized within the snails at the Pinturas River, for their part, presented morphologic similarities to a morphotype of *B. peregrina* registered in Minas Gerais, Brazil; but that Brazilian variant, in contrast, was demonstrated experimentally to be not susceptible to parasitism by *S. mansoni* (*Paraense & Deslandes, 1956*). Thus, determinations of this nature are fundamental for delineating the potential area of occupation of *B. peregrina* and obtaining a more realistic approximation of the corresponding potential zone of occupation of populations susceptible to parasitism, which information could be associated with the possible appearance of foci of infection and a southward dispersion of the endemium. In addition, regions with a moderate to high habitat suitability for *B. peregrina* were predicted within the Andes (Peru, Ecuador, Colombia, and Venezuela), which indicates a preference for relatively cold climatic ranges in that area and could have also implications on the possible risks of schistosomiasis in some specific altitudinal regions in the Andes.

Finally, as mentioned above, further research is still needed for a better understanding of the evolutionary history, ecology, parasitic susceptibility, and genetic variarion among the potential IHs of *S. mansoni* in South America, such as *B. peregrina*; which species is comparatively underrepresented in current research on planorbid snails despite its wide distribution in South America, as indicated by the evidence of the record of the new population described here in southern Patagonia.

## ACKNOWLEDGEMENTS

The authors would like to thank Dr. Monika Hamann (Centro de Ecología Aplicada del Litoral, CONICET) and three anonymous reviewers for providing valuable comments on an early version of the manuscript, and Patricia Sarmiento (Museo de La Plata) who obtained the scanning photographs. Finally, the authors are grateful to Dr. Donald F. Haggerty, a retired academic career investigator and native English speaker, for translating the manuscript from the original Spanish and editing the final version.

### Funding

This study was financially supported by National Scientific and Technical Research Council –Argentina (CONICET, PIP 796), and Facultad de Ciencias Naturales y Museo, Universidad Nacional de La Plata (Programa Nacional de Incentivos Docentes, N727). The funders had no role in study design, data collection and analysis, decision to publish, or preparation of the manuscript.

## Grant Disclosures

The following grant information was disclosed by the authors:
National Scientific and Technical Research Council–Argentina: CONICET, PIP 796.
Programa Nacional de Incentivos Docentes: N727.

## Competing Interests

The authors declare there are no competing interests.

## Author Contributions

- Alejandra Rumi, Roberto Eugenio Vogler and Ariel Aníbal Beltramino conceived and designed the experiments, performed the experiments, analyzed the data, contributed reagents/materials/analysis tools, wrote the paper, prepared figures and/or tables, reviewed drafts of the paper.

## DNA Deposition

The following information was supplied regarding the deposition of DNA sequences:
DNA sequences are available in GenBank under accession numbers KY124272–KY124273.

## Data Availability

The specimens are deposited in the malacological collections at the Museo de La Plata, Facultad de Ciencias Naturales y Museo, Universidad Nacional de La Plata, Argentina (MLP-Ma No 14186).

## Supplemental Information

Supplemental information for this article can be found online at http://dx.doi.org/10.7717/peerj.3401#supplemental-information.

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
