# Peer review of "The South-American distribution and southernmost record of Biomphalaria peregrina—a potential intermediate host of schistosomiasis"

_PeerJ, doi:10.7717/peerj.3401_

## Round 0.1 · original submission · Major Revisions

After careful consideration, we feel that the manuscript has merit and adds important information to the field. However, it does not fully meet PeerJ publication criteria as it currently stands. The three reviewers raise some very important concerns that must be addressed before this manuscript can be considered further. Some methodological concerns mainly those related to the limited number of sampled individuals analysed may have implications for your results and conclusions and must be considered. Please consider implementing all of the suggested changes by the three reviewers and address each of the reviewers concerns in a point-by-point response for resubmission.

Reviewer 1 ·

Basic reporting

No comment

Experimental design

Biomphalaria peregrina is not important for schistosomiasis. It never was found naturally infected. Besides, it is unclear to me why the 16S sequences was so short!!! This may have compromised the genetic analysis. About the phylogenetic analysis, it is more informative if the authors used more species from Biomphalaria genus to reconstruct phylogenetic history between species. In this study, the authors added two other species, whose role was like a outgroup.

Validity of the findings

The authors confirm the recording and the distribution of B. peregrina by robust techniques (molecular and morphological). However, phylogenetic analysis was poor, because the number of species.

Additional comments

The paper “The South-American distribution, southernmost record, and genetic variability of Biomphalaria peregrina – a potential intermediate host of schistosomiasis” presents a study about the B. peregrina specie in South-American. The work was well done. This manuscript describes the use of molecular and morphological assays to identify B. peregrina specie and its occurrence in South-American.
I am sorry that I cannot recommend this work for publication. However, I do encourage you to continue this study. My advice is to prepare a broader phylogenetic study of a larger sample of worldwide Biomphalaria, using additional markers.

A minor typo error: Page 264: change Mina Gerais to Minas Gerais and page 360: change Angentina to Argentine.

Reviewer 2 ·

Basic reporting

In the manuscript id.15126, by Alejandra Rumi and collaborators, authors report the Southernmost occurrence of the freshwater mollusc species Biomphalaria peregrina, a potential intermediate host for Schistosoma mansoni, in South America. They confirm the species identity through morpho-anatomical and molecular approaches. Authors also provide an up-to-date distribution model of the species based on their observations together with some occurrences previously reported in the literature. They discuss the distribution of this species -as well as their potential parasites- according to abiotic (i.e. climatic) and biotic (interactions with other snail species from the same genus) environmental factors and highlight the need for more studies on their model species.
The scope of the manuscript is clearly identified, the methods used are also described enough to be repeatable with only few minor issues (see details below). The introduction is well grounded in general ecological context and well document (although attention should be paid to the first paragraph (see details below). The result section is also generally clear as well as the tables and the figures that are of high quality. To my point of view, the discussion section has several issues. In particular, most of the discussion is speculative about several aspects that are not directly inferable from the results. In particular, authors discuss about (i) the possible interactions between B. peregrina and other snail species, (ii) the possible variation of compatibility between S. mansoni and B. peregrina from the two distinct major clades (i.e. tropical and temperate) or (iii) the possible effects of climatic factors limiting the propagation of schistosomiasis southward in South America. All these features, although of great interest cannot be inferred from the results obtained by the authors (see below for further details). On the other hand, the limitations of the study in regard with the methods were not discussed and are missing in the discussion.

Experimental design

This study is based on (i) the analysis of individuals collected from the field and (ii) a spatial distribution model based on the occurrence obtained from the study together with data collected in the litterature. Analyses are relevant and well conducted. However, the number of samples studied is very limited hence weakening the conclusions drawn by the authors (see below for further details).

Validity of the findings

Two main factors clearly weaken this study. First, this study mostly relies on the sampling of only 19 individuals collected from a single locality, among which only 12 individuals were morpho-anatomically analysed and only 5 were genetically characterised. Although this is likely to be sufficient to identify the species to which the snails collected belong, it seems difficult to draw further conclusions from this limited dataset, in particular concerning the “pattern of genetic variation of B. Peregrina” or the “evolutionary relationship of these southernmost individuals”. Second, although the distribution model of B. peregrina is an informative result, the conclusion from these analyses to infer the coldest temperatures as the “climatic barrier for the spread of schistosomiasis into temperate region” appear too speculative.
Below I detail some major issues concerning the discussion. I also address some minor issues that I hope will help to improve the manuscript.

Major comments on the discussion:
The discussion of the manuscript is organised into 6 major sections :
The first section consists in discussing the validation of the determination of sampled individuals as B. peregrina based on the results obtained on morphological and genetic analyses. This section is well written and clear.
In the second section authors discuss the factors that might explain the apparently very low intra-populational genetic diversity observed based on the 5 individuals analysed. As already mentioned, it is hard to draw some conclusions about genetic variability from only 5 individuals collected on the same site. One striking result however is that, according to Figures 2 and 3, this single haplotype seems to be shared with some populations geographically distant from the studied site (i.e. Agua escondida, roughly around 1000 km (?), although there is no geographical scale on the map). There are two main (non-exclusive) explanations to explain such pattern. (i) The resolution of the CO1 sequence used is not resolutive enough to detect genetic variability and discriminate these samples. In this regard it is most likely the case for the 269 bp 16S fragment used in this study. (ii) These populations share a close common evolutionary history. The founder effect hypothesis suggested by the authors only holds if the same haplotype spread and establish itself in two sites distant from several hundreds of kilometres which do not appear to be the most parsimonious hypothesis.
In the third section, authors discuss about the distribution of B. peregrina according to their results obtained from their distribution model. Here again, the section is well written although some minor changes could improve its clarity (see minor comments below).
The fourth section is dedicated to possible biotic factors that drive the geographical distribution of B. peregrina. More specifically authors discuss about possible biotic interactions between B. peregrina and other snail species from the same genus and more specifically the possible “competitive exclusion or displacement among them”. To my point of view this section is very speculative since no results obtained from this study could inform on the possible interactions between B. peregrina and other snail species. A comparison of geographical model distribution between species and the quantification of overlaps between these distributions together with experimental analyses would be necessary to infer possible interactions. I suggest to restrict this section to one or two sentences only or to remove it entirely.
In the fifth and last section, authors discuss about compatibility between S. Mansoni and B. peregrina and the possibility that the distribution of schistosomiasis in South America is limited by abiotic factors (i.e. cold temperatures) that influence free-living parasitic stages rather than biotic factors (i.e. presence/absence of intermediate hosts). The fifth section appears to be very descriptive and not necessarily based on authors results. As such I suggest that section to be shortened. The last section is mostly developed around other schistosomatidae species which is somewhat confusing. In ageement with the title of the manuscript as well as the introduction, it was expected that authors discuss about the possible distribution of S. mansoni rather than other schistosomatidae. One elegant manner to confirm the hypothesis that temperatures are expected to provide a climatic barrier to the expansion of schistosomes southward we be to build a distribution model of schistosomes based on occurrence data found in the littérature, and compare the distribution models between hosts (i.e. B. peregrina) and schistosomes. Without these complementary analyses, authors conclusions about the restricting factors that limit the distribution of schistosomes remain purely speculative.

Additional comments

Minor comments:

- The title does not reflect the content of the study. In particular the genetic variability aspect is exaggerated since only 5 sequences from only 1 single site were obtained and compared to pre-existing datasets with very little genetic resolution.
- Keywords: I would remove “genetic variation”

Abstract:
- Line 23: “although not found in the field”. Do you mean “although not found infected in the field”?
- Line 27: What do you mean by “displacement”? I would suggest “reassessment” rather than “displacement”.
- Line 30: I’m not sure that the present study was designed to assess “the pattern of genetic variation of B. Peregrina”. Authors have sequenced only 5 individuals which is not enough to determine the genetic variation within a population. Authors rather compared the genetic background of their samples with previously genetically characterised strains from S. America. This needs to be clarified.

Introduction:
- Schistosomiasis is a very general term to define diseases transmitted by schistosomes and can be declined into several kinds of schistosomiasis (e.g. intestinal, urogenital) depending on the schistosome species incriminated. Although Schistosomiasis broadly speaking indeed unfortunately affects more than 250 million people worldwide, schistosomiasis transmitted by S. mansoni is less severe. Authors should be careful and more accurate in their statements from line 51 to 60.
- A reference associated to the sentence line 61 to 63 is required.
- Too many references are associated with the idea developed line 84 to 88. Authors should focus on 3 or 4 key references.
- A reference associated to the sentence line 89 to 91 is required.

Material and methods:
- Line 114: The number of adult specimens analysed is required here.
- Line 115: information about the other anatomical features is required.
- Line I48: don’t understand how the consensus sequences were obtained. Do you mean consensus between forward and reverse sequencing?
- Line 182-184: I guess all South-American continental countries are enumerated in the list. Authors should consider removing this list.
- Line 186-187: “When the coordinates are lacking”. Given that most occurrences are prior to the 1980’s, there are certainly few GPS coordinates available. This limitation should be addressed in the discussion section since the precision of GPS coordinates used may influence the distribution model.

Results:
- Line 213: There is a mistake here: how could the mean be lower than the smaller diameter ?
Why n = 11 (instead of 12 for other traits) for the smaller diameter ?
- Line 215-221: The paragraph concerning the Radula is not clear. Sentences need to be clarified. There is no verb in the first sentence (Line 215 to 216). The following sentence (l. 216-217) is confusing. Please rewrite.
- Line 231: “six haplotype” should be changed to “six unique haplotypes”
- Line 245 to 247: These regions correspond to specific altitude areas within the Andes Mountain range. This clearly indicate that this species prefer relatively cold climatic ranges. This result also has impact on the possible risks of schistosomiasis in some specific altitudinal regions in the Andes. This should be mentioned and discussed.

Discussion:
- Line 270-272: Authors should tone-down their conclusions regarding the intrapopulational sequence variation since only 5 individuals were analysed. Authors should rather mentioned that “a single haplotype was found among the five individuals analysed”.
- Line 300-301 “the distribution”: The expected ? / The currently known ? / the characterised ?/ the potential distribution?.
- Line 311-315: This sentence is too long and somewhat confusing. Please rewrite.
- Line 315: “For this reason”: There is no clear causality between the two sentences. I suggest to rewrite this part.
- Line 324: “-it only a potential host”: This sentence is unclear, please rewrite.
- Line 340: Consider changing “in Results” to “in the Result section”.
- Line 350: Please indicate for which species these snails are intermediate hosts.
- Line 360 – 361: “to not be susceptible”: The structure of this sentence is a bit awkward. Please consider rewriting.
- Line 366: “With respect to region”: Do you rather meant “with respect to schistosomes” or “with respect to parasites” ?
- Line 369-370: Authors should refer to “free-living larval stages of schistosomes associated with B. peregrina” rather than “schistosome cercariae”. Molluscs are hosts for schistosome parasites or at best for schistosome miracidia but not for cercariae which are the stages emitted by the molluscs that infect the definitive host. In this regard, the sentence (as well as the sentence line 375) is awkward.
- Line 389: “research” rather than “reserch”

References:
Line 405: “and” is repeated twice
Line 496: “Larval” rather than “Laval”

Figures:
In Figure 2, is there a reason why the haplotypes are ordered as such ? (H4, H5, H6, H1, H3, H2). Authors should consider renaming them so as they appear along a more intuitive ordering (i.e. H1, H2, H3, H4, H5, H6).

Reviewer 3 ·

Basic reporting

The manuscript is technically sound, and the data support the conclusions. In my opinion, the manuscript presented in an intelligible fashion and written in standard English.

Experimental design

the analyses were suitable and minor revisions are necessary

Validity of the findings

I think this is an interesting manuscript. I am satisfied with several parts of it, in special the choice of methods and the species used. In my opinion, some aspects should be considered.

Additional comments

Lane 69 and 70 – …. include the name of the authors of the species B. peregrina (Orbigny, 1835), B. oligoza Paraense, 1974

Lane 72 - of shell morphology and the reproductive system ….include radula and jaw. Rumi (1991) used radula and jaw and the radula and jaw were used here (lanes 115 and 116).
Lane 74 - Estrada et al. 2006 …..include references about similar species (Vidigal et al. 2000, Spatz et al. 2000) as B. peregrina and B. orbignyi, B. oligoza found in Argentina.
Standley et al. 2011 mentioned that “The internal morphology of B. peregrina has been shown to distinguish it even from closely related species, but not from B. orbignyi (Spatz et al. 2000)”, please check and include.
Lane 74 – I would not entirely agree that these sentences were necessary, please check. In my opinion, these sentences about the doubtfull taxonomic position of the genus could be excluded (lanes 74-84, inclusive the references). I suggest that details about the morphologically similar species found in Argentina be included.
Lane 91– of the most widespread distribution…..from Venezuela to northern Patagonia - Include references describing B.peregrina distribution (Paraense, 1966, 2003, 2004, Standley et al. 2011, Nunez et al. 2010). Spatz et al. 2000 showed the presence of B. peregrina in others places from Argentina.
Include a sentence about the origin to its scientific name (from the latin peregrinus, meaning wanderer, in reference to its wide geographical distribution” (Paraense 1975., pg 117).
Lane 99 – genetic variation with available data for B. peregrina from GenBank. The majority of the references showed in table 1 are the unpublished data and the morphological studies were not showed. please clarify this point.
Lane 111 – include voucher numbers of the individuals used.
Lane 114 – In my opinion, should be included references which helpful the morphological identification of B. peregrina. I suggest the inclusion here of Paraense & Deslandes 1956, Paraense WL 1975b. Biomphalaria orbignyi sp. n. from Argentina Rev Brasil Biol 35: 211-222) and B. oligoza. Paraense WL 1974. Biomphalaria oligoza for Tropicobis philippianus (Dunker) sensu Lucena. Rev Brasil Biol 34: 379-386.

Lane 117 – Include the Raillet-Henry information according
Lane 120 - include ….following a modification of the non-destructive method described by Holznagel 1998 and Vogler et al. 2016 (Insights into the Evolutionary History of an Extinct South American Freshwater Snail Based on Historical DNA).
Lane 129 – include reference to radula formula (radula teeth Paraense & Deslandes 1956)
Lane 132 - include voucher number of the individuals used to molecular analysis.
Lane 151 – in order the confirm the morphology –based identification,…..please clarify. I suggest mentioned that sequences of the other similar species to B. peregrina were not used in the analysis.

Standley et al. 2011 16S COI Show reproductive system of B. peregrina
Collado et al. 2011 COI does not show reproductive system of B. peregrina
Collado & Mendez 2012 16S rRNA gene and ITS1 and ITS2 do not show reproductive system of B. peregrina
Tuan et al. 2013, Palasio & Tuan 2013 and 2016 COI 3 sequences of 16S unpublished data, morphological studies are not showed
Dejong et al. 2001 16S published data

Lane 155 - See table 1.

COI – 14 sequences were used here - 1 sequence was produced in this study, 4 were produced in published data and 9 were obtained from unpublished data.
16 S - 11 sequences were used here - 1 sequence was produced in this study and only two were obtained from published data. Please, clarify this point.

Which was the criterion of selection of the sequences chosen for the analysis?

Lane 209 - The features correspond to the standard description of the species ….., please clarify.
Lane 215 - The features correspond to the standard description of the species ….., please clarify.
Lane 223 - I would like to see the reproductive system of B. peregrina produced in this study. Please, check the possibility of include this figure in this paper.

Lane 260 – Discussion
I suggest that issues about the similarity among Bimphalaria species as B. peregrina which is the focus of this paper should be included here in the discussion. In my opinion, this point is important in the context of this study

Spatz et al. 2000 - “B. orbignyi Paraense, 1975 cannot be differentiated from B. peregrina neither by shell characteristics nor by the morphology of certain genital organs and has also been considered as a variety of B. peregrina by Orbigny (1837), as mentioned by Paraense (1975b). This species was originally described in 25 localities from Argentina and was described as refractary to S. mansoni infection (Paraense 1975b)”
Lane 272 – is important considered that others similar Biomphalaria species (found in Argentina) are not included in the analysis produced by Satandley et al. (2011). Please clarify this point here in the discussion. Is the COI region appropriate to separate similar species as Biomphalaria orbignyi, B. oligoza and B. peregrina? Please clarify
Lane 286 –include questions about the relationship with other similar species as reported by early molecular studies (Spatz et al. 2000, Vidigal et al. Phylogenetic relationships among Brazilian Biomphalaria)
“To verify whether the others molecular markers sequence analysis corroborates the classical morphological identification of the B. peregrina and showed the relationship among similar species”.

Lane 584 - include Paraense 1975a and Paraense WL 1975b. Biomphalaria orbignyi sp. n. from Argentina (Gastropoda: Basommatophora: Planorbidae). Rev Brasil Biol 35: 211-222

---

## Round 0.2 · accepted · Accept

All of the comments and points raised by the reviewers were properly addressed.

Reviewer 2 ·

Basic reporting

I've previously reviewed an early version of the manuscript. Authors addressed adequately all my comments. I consider that the manuscript can now be accepted for publication in PeerJ.

Experimental design

no comment

Validity of the findings

no comment

Additional comments

no comment

Reviewer 3 ·

Basic reporting

The manuscript was modified taking into account the majority of the referee´s suggestions. I am satisfied.

Experimental design

The manuscript was modified taking into account the majority of the referee´s suggestions. I am satisfied.

Validity of the findings

The manuscript was modified taking into account the majority of the referee´s suggestions. I am satisfied.

Additional comments

The manuscript was modified taking into account the majority of the referee´s suggestions. I am satisfied.